# Soil moisture-evaporation coupling shifts into new gears under increasing $CO_2$

Hsin Hsu [1] ✉ & Paul A. Dirmeyer [1,2]

When soil moisture (SM) content falls within a transitional regime between dry and wet conditions, it controls evaporation, affecting atmospheric heat and humidity. Accordingly, different SM regimes correspond to different gears of land-atmosphere coupling, affecting climate. Determining patterns of SM regimes and their future evolution is imperative. Here, we examine global SM regime distributions from ten climate models. Under increasing $CO_2$, the range of SM extends into unprecedented coupling regimes in many locations. Solely wet regime areas decline globally by 15.9%, while transitional regimes emerge in currently humid areas of the tropics and high latitudes. Many semiarid regions spend more days in the transitional regime and fewer in the dry regime. These imply that a larger fraction of the world will evolve to experience multiple gears of land-atmosphere coupling, with the strongly coupled transitional regime expanding the most. This could amplify future climate sensitivity to land-atmosphere feedbacks and land management.

Soil moisture (SM) variability can control the water and heat fluxes at the land surface, affecting air temperature and humidity[1–4]. This can establish feedbacks wherein, as the soil gets drier, the synchronous decrease in latent heat flux (LE, the energy used for evaporation) and increase in sensible heat flux (H)[5,6] results in a warmer and drier overlying atmosphere[7–9]. Increasing SM can set opposite changes into motion —each chain of events ultimately alters cloud formation and precipitation[10–15]. Such positive local feedbacks play an important role in extreme events such as droughts, floods, and heatwaves[6,16–20]. Furthermore, heterogeneous SM patterns that induce heterogeneous atmospheric heating and moistening can modulate mesoscale circulations and precipitation patterns[21–28], triggering or maintaining mesoscale convective systems (MCSs)[29,30], and propagating drought events[31]. All of these phenomena are rooted in the physical linkage between surface heat fluxes and SM, the foundation of SM-induced feedback.

SM-induced feedback can be strengthened or suspended in certain conditions. Such a shift in gears happens when surface fluxes disconnect from SM control under specific conditions[32–36]. Such a disconnection is related to important SM thresholds (Fig. 1): the wilting point (WP) and the critical soil moisture (CSM). WP is a criterion of vegetation hydraulic pressure that determines whether osmosis happens. When soils are drier than WP, water is unable to enter plant roots

and supply transpiration, the main evaporation component contributing to LE. Consequently, the sensitivity of evaporation to SM variability drops when SM < WP. This is called the dry regime. CSM is the threshold separating causes of evaporation limitation: water availability versus energy availability[3]. When SM > CSM, SM does not regulate evaporation[37,38]. Evaporation may even decline, as very wet soils correspond to periods of rainfall, cloudiness, and limited sunshine. In this wet regime, LE is also insensitive to SM variations. Only when SM is between WP and CSM, lying in the transitional regime, does LE reliably increase with increasing SM. The dry and transitional regimes together can be summarized as moisture-limited, while the wet regime is energy-limited.

Transitions among these three SM regimes have recently been shown to have a potential influence on extremes. The 2010/11 flooding in northern Australia moistened the land surface beyond the usual moisture-limited regime and decoupled land from atmosphere[39]. Reduced evaporation led to wetter land conditions that could maintain the flood. On the contrary, as SM dries, passing from the transitional regime to the dry regime, the available energy for surface heat fluxes goes mostly into H, exacerbating near-surface atmospheric heating. Consequently, air temperature becomes hypersensitive to declining SM, as has been evidenced recently in the United States[40] and Europe[41].

[1]George Mason University, Fairfax, VA, USA. [2]Center for Ocean-Land-Atmosphere Studies, George Mason University, Fairfax, VA, USA.
✉e-mail: hhsu@gmu.edu

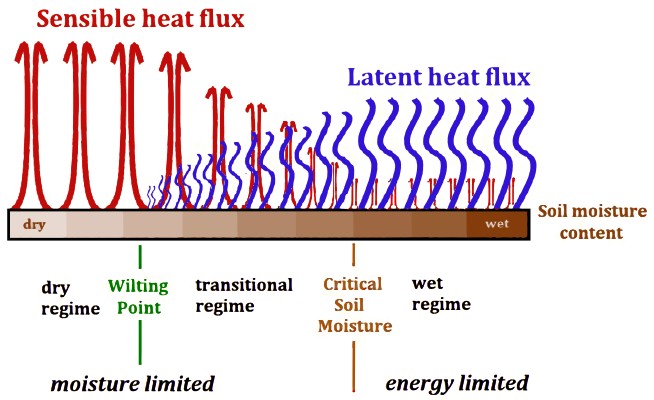

**Fig. 1 | The influence of soil moisture on the partitioning of surface heat fluxes with a fixed amount of available energy.** The full range of soil moisture (SM) can be separated into dry, transitional, and wet regimes by the thresholds of SM (wilting point WP and critical soil moisture CSM) or can be separated as moisture-limited conditions and energy-limited conditions separated by CSM. The transitional regime between WP and CSM is where latent heat (LE; the energy that supplies evaporation) and sensible heat (H) have a strong sensitivity to SM variations. Given a fixed amount of available energy (net radiation minus ground heat flux), variations in LE and H are shown as a function of SM (increasing from left to right) by the vertical lengths of the blue (evaporative flux) and the red (thermal exchange) symbols, respectively. The shutdown of LE when SM falls into the dry regime leads to available energy exclusively supplying sensible heat. In the wet regime, LE ceases to increase with increasing SM.

Accordingly, SM regime transitions are good indicators of the shifting gears between modes of local SM:LE coupling that are an essential component of SM-induced feedback. Though an improved understanding of how SM-induced feedback can change under global warming has been a priority for some time[3], studies have focused more on quantifying the change of coupling in a climatological sense, such as a projected strengthening in land–atmosphere[9,42–46] or an expansion in areas experiencing strong coupling[47,48]. However, the climate in many locations is composed of days with and without active SM:LE coupling and thus the existence of SM-induced feedbacks. This suggests that under global warming, in addition to stronger control of SM on LE, the climatological strengthening in coupling can also be attributed to a more frequent control of SM on LE, i.e., more days when SM is between WP and CSM, or even the situation where a transitional SM regime emerged locally when it did not exist previously. Thus, investigation on whether a warming climate leads to the shift, emergence, or disappearance of SM regimes, along with corresponding changes in the frequency of SM in each SM regime, has been lacking but is needed.

In this study, we determine the global patterns of existing SM regimes and their projected changes from state-of-the-art climate models. This enables quantification of how SM values migrate among dry, transitional, and wet regimes due to global warming. Such diagnostic analyses can indicate which gear of SM-induced feedback is dominant at any location, and what changes may occur. To examine these responses under warming with a climate model consensus perspective, daily data from ten climate models participating in the

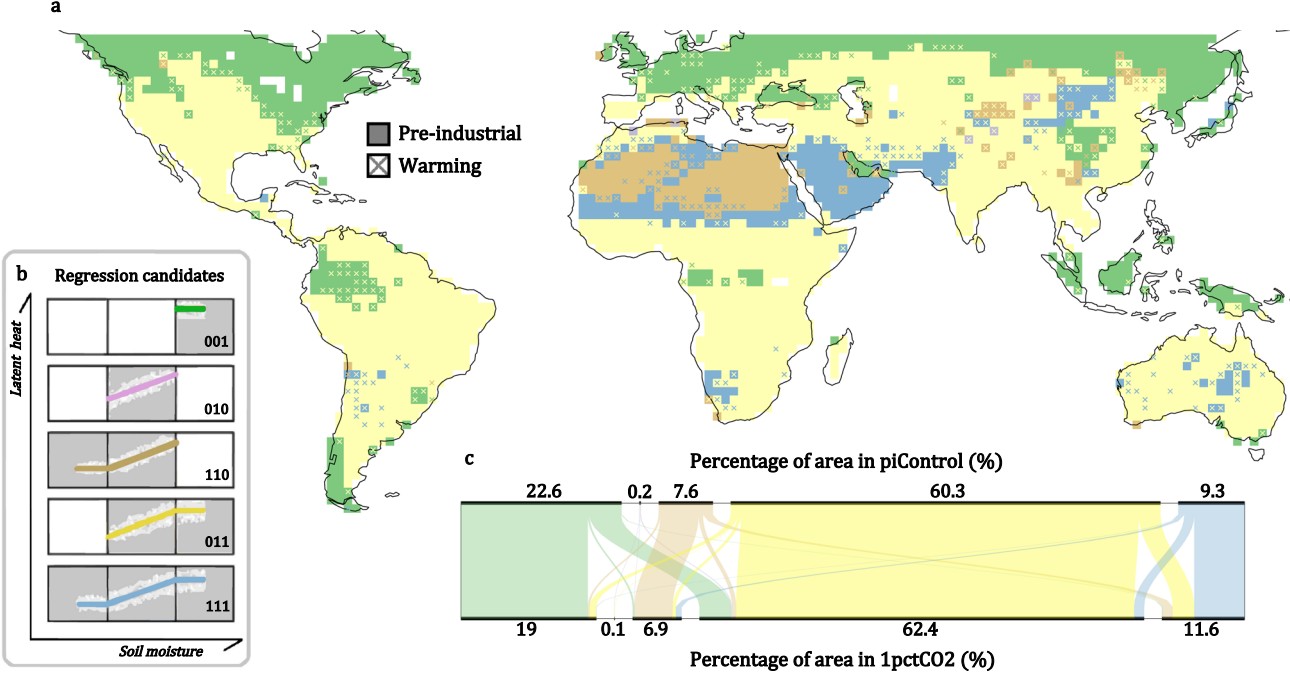

**Fig. 2 | Global distribution of soil moisture regimes and their shifts under global warming.** Soil moisture (SM) regimes at each grid cell determined by the multi-model mode of the selected segmented regression candidate (**a**) among the five depicted regressions (**b**): wet regime (a.k.a. energy-limited; green), transitional regime (purple), dry + transitional (a.k.a. moisture-limited; brown), transitional + wet (yellow), and dry + transitional + wet (blue). For the candidate schematics on the left, soil moisture (SM) increases along the x-axis, and the evaporation rate (LE) increases along the y-axis. In the global map (60°S–60°N), the color of each grid cell represents the elected candidate from the pre-industrial climate and the cross color (over the square color) indicates a new candidate emerging in a warming climate at that grid cell. The alluvial diagram (**c**) shows the shift in the coverage of each candidate, calculated as the percentage of the global land area between 60°S to 60°N. (Source data are provided as a Source Data file[78]).

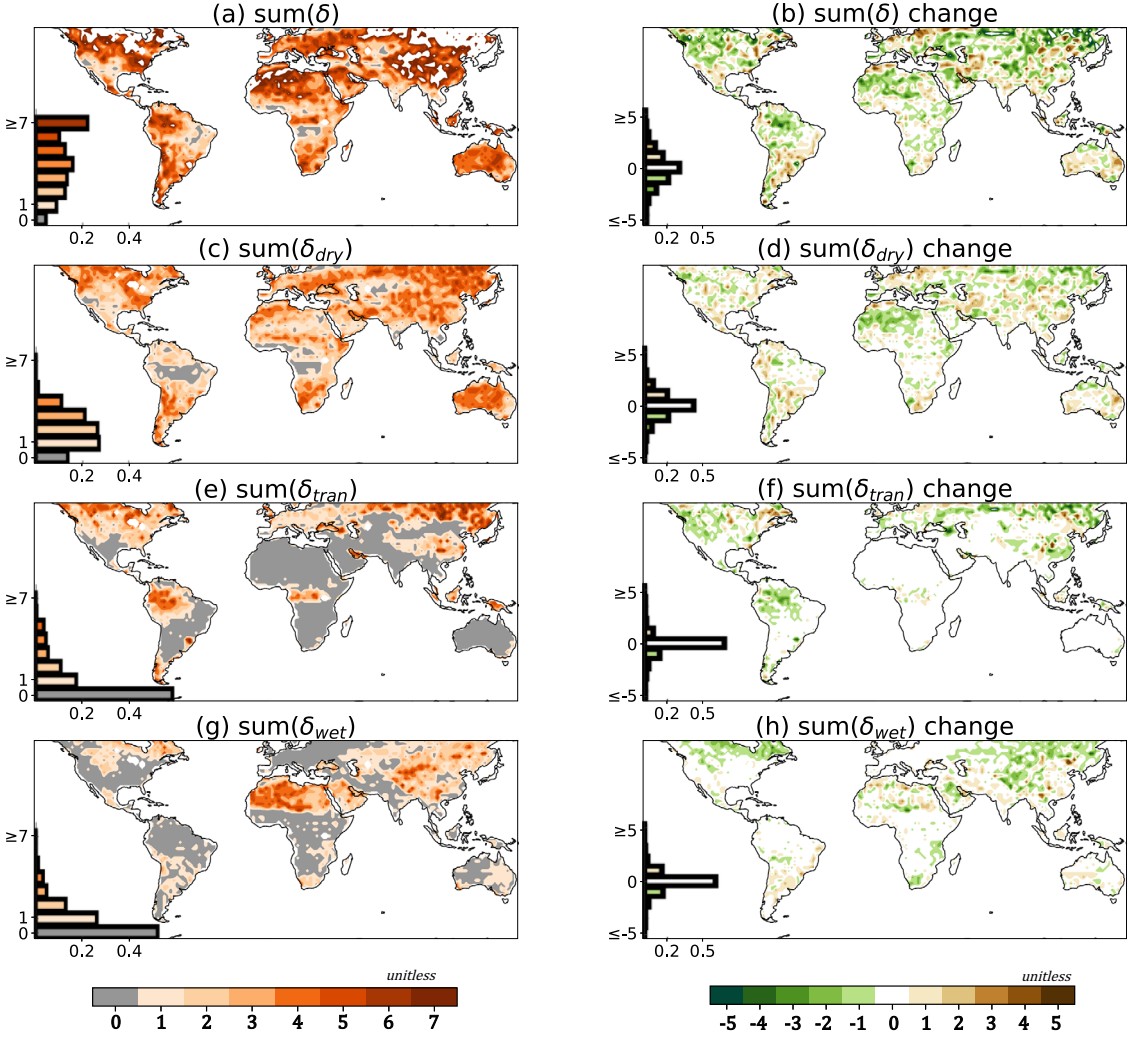

**Fig. 3 | The disparity of soil moisture regimes among climate models and their changes under global warming.** Sum($\delta$) quantifies the disparity among models in soil moisture (SM) regimes compared to the multi-model consensus (see Methods for formulation); a larger sum($\delta$) indicates greater disparity and, thus, the lower consensus. **a** Displays sum($\delta$) for piControl simulations. **b** Displays the sum($\delta$) change under global warming (1pctCO2 minus piControl). **c, e, g** Display the individual consensus of detection of the dry, transitional, and wet regimes, respectively. **d, f, h** Display the corresponding changes in the consensus under global warming. Histograms in each panel display the fractional land area (60°S to 60°N) having the specific value of sum($\delta$) or change in sum($\delta$). (Source data are provided as a Source Data file[78]).

Coupled Model Intercomparison Project Phase 6 (CMIP6) from simulations proxying pre-industrial climate and warming climate (Methods) is used.

## Results

### Pre-industrial and projected patterns of soil moisture regimes

Breakpoint analysis based on theoretical SM:LE behavior[49,50](see Methods) is used to find the two critical values, WP and CSM at each location, and determine which SM regimes are active there. A total of five candidates, representing different combinations of active SM regimes, are used to represent which regimes are active in each grid cell (Fig. 2b). We assign binary digits 0 and 1 to indicate the presence of the dry, transitional, and wet regime, in order, following the letter C for "candidate" (e.g., C010 for the transitional regime only). Figure 2 displays the spatial pattern of consensus SM regimes among the climate models and how they are projected to change under increasing CO$_2$. This is determined by the mode of the candidate selected among all climate models. The five colors correspond to each of the five candidates. Grid cells are colored according to the pre-industrial climate

model consensus, and crosses within grid cells indicate the projected candidate in a warming climate (crossed grid cells account for 13.9% of the global analyzed area). Note that the values of WP and CSM may also change (see Fig. S1 and Discussion).

Over tropical rainforests, candidates C001 and C011 dominate in the pre-industrial climate (Fig. 2a). Only for C011 can SM fall within the transitional regime in these locations, signifying active SM:LE coupling. This situation becomes more common under warming as C011 expands in area, particularly over the Amazon. Over semiarid regions such as the Sahel, Australia, and southern Great Plains, candidates C011 and C111 dominated during the pre-industrial period. C111 expands in a few of these regions under warming. In arid regions, SM does not always remain in the dry regime. C110 and C111 occupy the Sahara, Chile, and Arabia. This could be because rare precipitation events moisten the land surface enough to spark significant evapotranspiration that registers in our breakpoint analysis. SM surpasses CSM in more locations as C111 emerges under warming.

Overall, the global area where LE is sensitive to SM outside high latitudes (60°S–60°N) increases by 3.6% under global warming, as

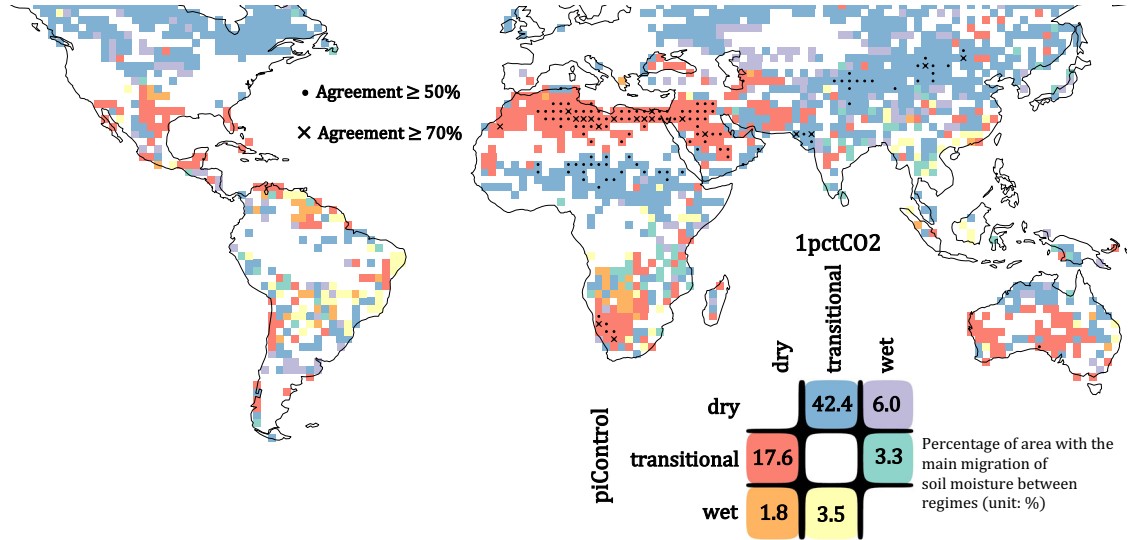

**Fig. 4 | The main migration of soil moisture frequency between regimes under global warming.** For each climate model, the change in the percentage of days (1pctCO2 minus piControl) that soil moisture (SM) is in each regime is calculated, and its significance is tested by a chi-square test of independence with $p < 0.05$. If no significant difference in at least one SM regime is found, that grid cell is classified as experiencing no migration (blank). If there is a significant difference in any SM regime, net migration is classified as a shift from the SM regime with the largest decrease in frequency to the regime that has the largest increase. The main migration is obtained by the mode among the climate models. Grid cells with an agreement of at least 5 of 10 models are marked with dots; agreements of at least 7 of 10 models are marked with crosses. The color key also displays the percentage of land area (60°S–60°N) experiencing each migration in a warming climate (24.9% of the global land area shows no migration). (Source data are provided as a Source Data file[78]).

indicated by the summation of candidates for which the transitional regime is active, shown in the alluvial diagram in Fig. 2c. Only 22.6% of global land areas are identified as uncoupled (C001) for pre-industrial climate; a 3.6% expansion in a global area with active SM:LE amounts to shrinkage of 15.9% in the uncoupled area.

## Model agreement regarding soil moisture regimes

The metric sum($\delta$), previously defined[50] (See Methods), is used to quantify the disparity among climate model simulations of SM regimes for the pre-industrial climate (Fig. 3) and how these disparities change under global warming. Similarly, its decomposition into sum($\delta_{dry}$), sum($\delta_{tran}$), and sum($\delta_{wet}$) in Fig. 3 reflect the degree of model disparity in simulating dry, transitional, and wet regimes individually.

The strong disparity in representing the SM transitional regime is found over rainforests (Fig. 3e), especially over the Amazon, and this disagreement is largely reconciled in a warming world (Fig. 3f). The dry regime is detected in a few climate models around the edge of the Amazon (Fig. 3c). The consensus is relatively high in semiarid regions compared to the rest of the world (Fig. 3a). Disagreement is mainly found for dry regime detection (Fig. 3c). Note that some climate models do not simulate a dry regime over the Sahara, as sum($\delta_{dry}$) is not zero (Fig. 3b). Over this area, detection of the wet regime is also inconsistent across models (Fig. 3g) and the consensus is even less under warming (Fig. 3h).

The fractions of the area with a change in any specific value of sum($\delta$) and their changes are displayed in the histogram of each panel. Disagreement regarding SM regimes is mainly contributed by sum($\delta_{dry}$). The overall consensus of modeled SM regimes increases under warming, due mainly to better agreement in the detection of transitional and wet regimes.

## Main migration tendency for soil moisture between regimes under global warming

A locally emergent SM regime does not necessarily indicate that SM is predominantly migrating into that specific regime (see Table. S1 for a local example). We examine the net migration of SM between regimes under global warming (see Methods). This yields seven possible categories: no migration of SM or a migration of SM between any two of the three defined SM regimes. The results are shown in Fig. 4, with symbols indicating different levels of agreement. The fraction of land area (60°S–60°N) within each category is displayed using the color matrix.

The tendency of SM to extend into the transitional regime grows in a warmer climate over some semiarid regions. Individual model changes are displayed in Fig. S3. Spread among climate models at any location can be large. This is mainly due to differences among models in the detection of WP and/or CSM. Nevertheless, the trend of how SM migrates in a changing climate demonstrates strong agreement in many regions. Over northwestern India, the Sahel, northern Australia, and central Asia, SM shifts from the dry regime to the transitional regime, as suggested by most climate models. On the other hand, over southern Africa, western and central Australia, migration is from the transitional regime to the dry regime, corresponding to the regions that C111 emerges under warming. Over arid regions, despite the strong disparity in detected SM regimes (Fig. 3), how SM migrates between the regimes under warming is consistent among the climate models (Fig. 4). In the Sahara, Chile, and Arabian Peninsula, SM lies more frequently in the dry regime, usually indicated by at least 7 of the 10 climate models. Migrations between the dry and transitional regimes account for 60% of the global area (red + blue) categories in the color matrix of Fig. 4) and migration from dry to transitional (blue) is the major tendency across the globe (42.4%). Robust results with at least 7 of 10 climate models agreeing on the same tendency are often found over arid regions and semiarid regions. Generally, no significant migration of SM between the regimes due to warming is shown over the deep tropics (Fig. 4). Sporadic responses are found over the Amazon and the maritime continent, while spatial patterns are not homogeneous and are not consistent among the climate models.

## Discussion

SM regimes are a good determinant for the type of local land–atmosphere coupling, which is determined by the relationships between surface heat fluxes and SM (Fig. 1). Under global warming, SM

can extend to unprecedented SM regimes in many locations (Fig. 2). There is a trend in many locations toward more gears of SM:LE coupling being experienced, and thus more shifting between gears, suggesting less stable hydroclimates. Specifically, more locations experience the transitional regime while few locations show that an SM regime will vanish (Fig. 2). A consequence of this broadening of SM:LE candidates is that model consensus grows with increasing $CO_2$ (Fig. 3). For both the pre-industrial and the warming climate, SM spans at least two regimes over most of the world (Fig. 2). This suggests that evaluating the temporal variation of coupling is necessary when investigating topics relevant to land surface processes, especially for extreme events. Furthermore, the expanding transitional regime could amplify local climate sensitivity to land–atmosphere feedback. This could make some regions more susceptible to unintended consequences from land management practices that alter soil moisture, such as irrigation, cropping choices, urbanization, and water resource management.

The WP and CSM are well defined by the breakpoint analysis (see Methods); however, WP and especially CSM can be sensitive to ecological and environmental factors. Therefore, values estimated here act as a climatology of the varying breakpoints of each analyzed period. WP is mostly determined by soil and plant properties. For most climate models, plants are prescribed identically in piControl and 1pctCO2 simulations. This results in a negligible change in WP under increasing $CO_2$ (Fig. S1b). CSM is affected not only by vegetation characteristics[51] but also by climate conditions such as vapor pressure deficit and insolation[52–54]. As these conditions do change in a warming climate, there can be larger shifts in CSM under increasing $CO_2$, as seen in Fig. S1c. Accordingly, an emerging SM regime under global warming is often not completely attributable to a shift in SM distribution, but shifts in WP or CSM may also contribute. For instance, a transitional regime is seen to emerge over the Amazon. In addition to a drier SM distribution (Fig. S1a), there is also a trend toward higher values for CSM (Fig. S1c), which embrace a wider range of SM in the transitional regime, bolstering its emergence.

SM migration between regimes under global warming (Fig. 4) shows a zonally consistent pattern. Such a response of SM distribution under warming is presumably due to changes in the large-scale atmospheric circulation and/or regulated hydrological cycle[55–57]. SM migrates from the dry to transitional regimes over the Sahel, southern Arabian Peninsula, and Northern Australia. The increase in the number of days that SM actively controls LE might strengthen the SM-induced feedbacks in these regions, which have been long-recognized as "hot spots" of land–atmosphere interactions[58–65].

Meanwhile, SM regimes migrate from the transitional to dry regime over the Sahara, northern Arabian Peninsula, southern Africa, and western and central Australia. As a result, the arid regions spend more days in the dry SM regime, while the transitional semiarid to semi-humid regions spend more time in the transitional SM regime. Although this appears to correspond to the "wet gets wetter, dry gets drier" paradigm[66,67], results here do not necessarily suggest that transitional regions get wetter and arid regions get drier. As mentioned above, critical SM values can shift under a warmer climate, and thus parts of the SM spectrum can slide into different regimes without a significant change in the SM distribution. How precipitation, LE, and temperature individually affect the shifting gears of SM:LE coupling needs to be disentangled.

Regimes detected over higher latitudes (above 50°) or higher altitudes could be underrepresented, especially for the pre-industrial climate. Inactive SM:LE coupling is mainly attributed to snow cover that cuts off the connection between soil and atmosphere. For instance, C110 is suggested to best describe the SM distribution over eastern Siberia in some climate models (Fig. S2). This does not guarantee the detection of a dry regime. The subarctic climate there leads to notable moist and warm conditions during summer, driving evaporation. Winter is the dry season from a precipitation perspective; however, regardless of the dryness of SM, the snow cover stops the evaporation of soil moisture. As the number of days that land is covered by snow varies greatly among climate models, a less reliable result is likely over such areas (Fig. 3a, c, g). Nevertheless, a consistent pattern is seen under climate warming. In Fig. 2, C011 expands poleward over North America and Eurasia. With more snow-free days under warming conditions, the sensitivity of LE to SM emerges, and land–atmosphere interactions could become important in regions where they currently are not[42].

Some locations indicate different soil moisture regimes in different climate models within the same 50-year climate; such disparities can be large (Fig. 3). We argue that this disparity is not guaranteed to decline even if more climate models could be included in this analysis. The metric sum($\delta$) quantifies uncertainties that can be extended to characterize the fidelity of projections of extreme events linked to SM:LE coupling, such as heatwaves, in multi-model analysis. Figure 3 can help to indicate the usefulness of multi-model projections relevant to land surface processes at any location. Specifically, a better consensus is reached over semiarid regions (e.g., the Sahel, India, and Southern Great Plain), while discrepancies are relatively strong over the rest of the globe (Fig. 3a).

Few climate models simulate a dry regime over heatwave-active regions such as western North America (Fig 2 and Fig. S2). A similar pattern of biased SM dry regimes in historical simulations of climate models has been pointed out previously by comparison to observationally-constrained data sets using the same regime detection method[50], wherein the dry regime is more commonly indicated in North America, Europe, and Australia. However, here we have used pre-industrial simulations instead of historical simulations that correspond to the period of observational data sets, for reasons described in the Methods summary; the lack of SM dry regime is seen globally in all analyzed climate models (Fig. 2). As the dry regime is more prevalent in a warming climate (Fig. 3d), the lack of dry regime might be attributed to a wetter climate during the pre-industrial period, rather than a problematic parameterization that prevents soil wetness to fall below the WP. Furthermore, different strategies of how climate models represent phenology[68] can affect WP, introducing another uncertainty in the aggregated result of SM regime detection. Moreover, different land models that inherently simulate different soil moisture distributions[69], which also often differ from observations, remains an issue[50]. This leads to an essential discrepancy among preferred SM regimes in the climate models inducing uncertainty in the multi-model diagnoses and estimated migrations in this study. Regarding these issues, a stricter validation of models' SM regimes may become possible in the future, based on promising developments in global SM observations[70–74]. A more comprehensive analysis of each component of SM-induced feedback in climate models will further help to evaluate the credibility of this study and to understand the causes of bias and change of models' SM regimes.

The global maps provided here can suggest locations worthy of a regional study of the physics that drive SM:LE coupling and atmospheric responses. For example, SM lingers more in the transitional regime over the Sahel in a warming climate. Does this only indicate more days with active SM:LE coupling, or will this strengthen the magnitude of SM:LE coupling and thus affects long-term temperature climatology? Over Australia, SM that shifts into the dry regime in the north and into the transitional regime in the south could lead to an opposing shift of gears in SM-induced feedback. Investigating further the underlying mechanisms and how they impact extremes can help us to understand land–atmosphere interactions, potentially aid prediction and mitigation, and assess climate vulnerabilities with a new perspective.

## Methods

Ten climate models participating in CMIP6 are used: MIROC6, AWI-ESM-1-1-LR, CMCC-ESM2, CanESM5, CNRM-CM6-1-HR, NorESM2-MM, IPSL-CM6A-LR, MRI-ESM2-0, GFDL-CM4, and INM-CM4-8 (see Table S2). These models are selected because they provide daily SM and LE fields from DECK simulations[75] for both the piControl and 1pctCO2 cases (data are available online at: https://esgf-node.llnl.gov/search/cmip6/) at the time of our analysis. Daily SM (CMIP variable mrsos; soil water content in the top 10 cm) and LE (CMIP variable hfls) fields are taken from these simulations. The ensemble member r1i1p1f1 is used in all simulations except for CNRM-CM6-1-HR, where r1i1p1f2 is used.

To provide a robust indication of the shifts in SM regimes under warming, analysis is performed for the portion of 1pctCO2 runs spanning the 100th to 150th year, the time that the concentration of $CO_2$ passes from ~2.5 times the piControl level to 4 times, and is compared to the results from a 50-year period of the piControl run. (see Table S2 for the model years used) SM and LE fields from each model are regridded to a common $2° \times 2°$ global grid; data are interpolated from the nearest grid cell. We note that daily LE fields are available from more models than daily H fields, so that SM:LE is used here to determine SM regimes. The piControl simulation is used instead of the historical simulation as the baseline for comparison because it is more consistent with 1pctCO2 runs. The historical simulation includes forcings in addition to greenhouse gases, such as land cover change and aerosol variability, that complicate diagnoses; comparison between piControl and 1pctCO2 runs yields a clean assessment of the response to increasing $CO_2$.

### Soil moisture regime determination

Segmented regression[49,50] is used to define the two soil moisture thresholds (WP and CSM) as breakpoints in piecewise linear fits to data where SM is the independent variable. This analysis is performed for each model and simulation. At each grid cell, five different segmented regression candidates are fitted to the distribution of available total daily LE versus SM for each simulation by each climate model; these segmented regression candidates are determined as follows: Locally, at least one SM regime (dry, transitional, or wet regime), to a maximum of all three SM regimes, can be determined depending on whether either or both of the SM thresholds are detected. This yields six possible combinations of SM regimes, called candidates: a solely dry regime C100, a solely transitional regime C010, a solely wet regime C001, a dry + transitional regime C110, a transitional + wet regime C011, and a dry + transitional + wet regime C111.

A solely dry or wet regime with SM:LE dependency (candidates C100 or C001) is indicated by a one-segment regression with zero-slope. These are found to arise almost exclusively over rainforests and at high latitudes where soils are almost always wet. Rare cases can also be found at coastal regions dominated by maritime air. Thus, we treat all zero-slope cases as candidate C001. A solely transitional regime is indicated by a one-segment regression consisting of a segment with a positive slope (C010). Two segment regressions have one breakpoint. A dry + transitional regime is indicated by a two-segment regression consisting of a constant (dry) segment followed by a positive slope segment (C110). The transitional + wet regime is indicated by two-segment regression consisting of a positive slope segment followed by a constant (wet) segment (C011). A full dry + transitional + wet regime is indicated by the candidate with a three-segmented regression and two breakpoints, consisting of a positive slope connecting two constant segments (C111).

Bayesian information criterion (BIC) for statistic model selection[76] is used to select the best fit among the five segmented regression candidates:

$$BIC = n\ln(RSS/n) + k\ln(n) \qquad (1)$$

where $n$ is the sample size, $RSS$ is the residual sum of squares, and $k$ is the number of model parameters, which is used to penalize regressions with a more complex structure to prevent overfitting. At the same time, values of the WP and CSM, if detected by the best-fitted regression, are recorded at each grid cell. It is possible for WP or CSM to change between simulations for the same model, and this aspect is also investigated. Akaike information criterion was also tested and found to produce nearly identical results.

### Agreement of model soil moisture regimes

We have previously designed an index $\delta$ to count the number of SM regimes for which model detection disagrees between any two candidates[50].

$$\delta = |a - x| + |b - y| + |c - z| \qquad (2)$$

where $a$, $b$, and $c$ are the dry, transitional, and wet binary bit, respectively, of a candidate value (e.g., for a particular model); $x$, $y$, and $z$ represent the same digits as $a$, $b$, and $c$ for another source of candidate values (e.g., for validation data). For example, candidate 001 and candidate 110 have bitwise opposite detected regimes resulting in a maximal $\delta = 3$. Candidate 111 and candidate 110 only have a disagreement on the detection of the wet regime, and thus $\delta = 1$. By replacing $x$, $y$, and $z$ with the digits of the mode of the candidate obtained among all climate models, $\delta$ can represent how a specific climate model departs from the multi-model consensus. Accordingly, the summation of the $\delta$ values obtained for all pairs of each available climate model and the mode can represent how disparately the climate models simulate local SM regimes. Consequently, the lower the summation of $\delta$, the better the agreement and the stronger the consensus of SM regimes among the climate models.

### Migration of soil moisture among regimes

The fraction of days that SM stays in each SM regime is calculated for each simulation. The changes of these fractions are calculated for days that SM stays in the same regime between piControl and 1pctCO2 (Fig. S3). Then, the chi-square of independence test[77] is applied to determine if the change of the fraction is statistically significant. If a difference in the fraction of days for any of the three regimes is significant at the 95% confidence level, the regime with the largest decrease is tagged as the regime that SM shifts out of, while the regime with the largest increase is tagged as the regime that SM shifts into. These tags yield seven categories: no statistically significant migration of SM or a migration of SM between two of the three SM regimes. Finally, the mode of the category is obtained for each grid cell from the same grid cell of each simulation by each climate model.

We also provide the same analysis using total soil moisture content (CMIP variable mrso). Only 4 out of the 10 climate models (MRI-ESM-0, CNRM-CM6-1-HR, INM-CM4-8, and MIROC6) provide this variable for the same experiments and periods as for surface soil moisture data. These analyses suggest shrinkage in the solely wet regime area over the tropics (Fig. S4) and overall growth of the transitional regime (Fig. S5). Despite lower credibility due to fewer analyzed climate models and less clarity in the patterns of change, these results also indicate shifts in the dominant SM regime under global warming.

## Data availability

CMIP6 model data were available at: https://esgf-node.llnl.gov/search/cmip6/. Specific fields used in this work can be assessed by checking the boxes of the website with corresponding model names, experiments, time steps, and variables indicated in the Methods. The data generated in this study have been deposited in the repository under https://github.com/hhsu81819/Soil-moisture-regime-and-projection/tree/2.0v (https://doi.org/10.5281/zenodo.7586396).

## Code availability

The code for generating the results and plots of this study have been deposited in the repository under https://github.com/hhsu81819/Soil-moisture-regime-and-projection/tree/2.0v (https://doi.org/10.5281/zenodo.7586396). The alluvial diagram in Fig. 2 is generated by: https://www.mathworks.com/matlabcentral/fileexchange/66746-alluvial-flow-diagram.

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

## Acknowledgements

This work was supported by the National Aeronautics and Space Administration (NASA) grant 80NSSC20K1803.(PD received this funding)

## Author contributions

Both authors, H.H. and P.A.D., conceived of the presented idea, discussed the results, and contributed to the writing of the article. H.H. designed a computational framework performed the analysis and drafted the manuscript.

## Competing interests

The authors declare no competing interests
