## [Peer Review File · Nature Communications]

Soil moisture-evaporation coupling shifts into new gears under increasing CO₂REVIEWER COMMENTS

Reviewer #1 (Remarks to the Author):

Hsu and Dirmeyer classify evaporative regimes using several climate models and determine that land surfaces are on average projected to be in more water-limited evaporative states. I find this to be an important and timely study. It is methodologically sound in using previously developed methods to identify evaporative regimes and having proper confidence tests – I would have done all the same steps myself and the authors answered all of my methodology questions in their text. I support the study, but have some doubts about how well ESMs at the CMIP level, which tend to have very simplistic representations of the land surface, can really capture EF-soil moisture relationships and soil moisture trends that drive the main results. See my major comment for more details. I acknowledge it is difficult for the authors to address this concern, but I recommend some ways they may be able to provide some confidence in the model results.

I do not wish to remain anonymous – Andrew Feldman.

Major Comments

1) A major unanswered question and lynchpin of this study is how well CMIP climate models can really capture evaporation-soil moisture relationships as well as trends in soil moisture – both of which combine to give the results in Figure 4. Koster et al. 2006 in the GLACE experiments show a wide range of variability between models in this coupling that drives differences in the specific rain-soil moisture interactions they are looking at (which partly is a function of EF-soil moisture relationships). Short Gianotti et al. 2019 in their Figure 1 show a concerning wide range of variability of EF-soil moisture relationships in CMIP5 models that can often greatly differ from observations in the United States. Also see several studies by Jianzhi Dong and Wade Crow that show biases in land surface temperature – soil moisture coupling in models. Even though we can use an ensemble model average to communicate results here and find agreement (as in Figure 3), I am concerned with the high variance between models' EF-soil moisture relationships and L-A coupling (as shown in Koster et al. 2006 and Short Gianotti et al. 2019). Based on those studies, one could argue that we are still unsure how well these climate models are really capturing land-atmosphere interactions given their large variance in the EF-soil moisture relationships. EF-soil moisture relationships are emergent behavior from many parameterizations (for example, plant-soil-climate interactions defining WP and CSM which are still highly uncertain in these global models), each having their own errors and differences between models. Additionally, precipitation projections are yet uncertain with heterogeneous changes that translate to uncertain changes in soil moisture trends globally as well (see studies by Alexis Berg). Ultimately, Figure 4 aggregates all of these uncertainties into how much more time is being spent in a water-limited evaporative stage in a changing climate.

Unless the authors argue differently that there have been many previous studies supporting how well climate models capture land surface evaporative regimes and soil moisture trends, I highly recommend that the authors devote more text to addressing these issues as a main shortcoming of the study that it is still not well known how well CMIP/climate models capture land-atmosphere coupling and regimes. Personally, I think we need more explicit observational studies that can be used to assess how well climate models are capturing water-limited evaporative regimes (Both the shapes of EF-soil moisture relationships and time spent in the water-limited regime as two example metrics). There are unfortunately not many observational, model-free options that can be used to test how well the ensemble model classifications in Figure 2 and current climate mean time spent in the transitional regime are capturing reality.

Beyond explicitly commenting on these limitations, I recommend that the authors also find a way to build confidence in the ensemble model results for the present climate runs. A few recommendations: (1) FLUXNET EF-soil moisture relationships can offer

some sparse insights. (2) LST-soil moisture relationships from remote sensing (SMAP and MODIS) can give some larger spatial scale observational insights. (3) More careful selection of CMIP6 models by removing models that qualitatively show large deviations from expected nonlinear EF-soil moisture relationships in select test land domains (Figure 1 in Short Gianotti et al. 2019 can be a guide for this).

Short Gianotti, D.J., Rigden, A.J., Salvucci, G.D., Entekhabi, D., 2019. Satellite and Station Observations Demonstrate Water Availability's Effect on Continental-Scale Evaporative and Photosynthetic Land Surface Dynamics. *Water Resour. Res.* <https://doi.org/10.1029/2018WR023726>

2) I found the motivation to be lacking a bit here. The authors could do a lot more to explain why we should care that land surface processes are tending to become more water-limited.

Line Specific Comments

L55-57: Being rigorous, L55 "will not cause greater evaporation" and L57 "LE is also insensitive to SM variations" (and in caption in L625) are too definite of statements and may not be entirely correct. Soil moisture increases can still cause small evaporation increases in non-water limited regimes, but EF variations depend mainly on other factors.

Not required to reference but: Feldman et al. 2022 Fig. 6B+6C show with observations that there is still some energy flux sensitivity to soil moisture in the "wet regime" but it is just less. Dirmeyer et al. 2000 also shows this with models.

Dirmeyer, P.A., Zeng, F.J., Ducharne, A., Morrill, J.C., Koster, R.D., 2000. The Sensitivity of Surface Fluxes to Soil Water Content in Three Land Surface Schemes. *J. Hydrometeorol.* 1, 121–134. [https://doi.org/10.1175/1525-7541\(2000\)001<0121:TSOSFT>2.0.CO;2](https://doi.org/10.1175/1525-7541(2000)001<0121:TSOSFT>2.0.CO;2)

Feldman, A.F., Gianotti, D.J.S., Trigo, I.F., Salvucci, G.D., Entekhabi, D., 2022. Observed Landscape Responsiveness to Climate Forcing. *Water Resour. Res.* 58, e2021WR030316.

L71: A stronger, more explicit research question or objective is needed here. Stronger motivation is needed on why we should know the answer to this question and how well others have answered this previously. For example, Dirmeyer et al. 2012 referenced here have previously shown a projected increase in land atmosphere coupling strength with climate change which may or may not relate to the investigation here.

L75: Given my major comment, I think more motivation should be given for why the CMIP6 models are chosen. I am not a modeler so I'll word as a question: could we trust the analysis more if done with more land surface-focused models (CESM, CATCHMENT-CN) rather than ESM's that are coupled to a highly parametrized atmosphere and may have more complex land surface schemes (i.e., plant hydraulics) turned off?

Figure 2: As part of my comment below, it seems worth mentioning that there are very few pixels changing regime dominance based on this plot. 2.6% is very small. Perhaps shifts in the soil moisture distribution and specifically its mean are doing more of the work in changing time in the water-limited evaporative stage than changes in the tails of the soil moisture distribution that Figure 2 is probably more sensitive to.

L141: For clarification, am I supposed to interpret Figure 4 as showing more common migration from dry to transitional (green) than from transitional to dry (red)? This could be made clearer in the text here. The main message of the study can get a little lost when starting with Figure 2, where the changes in the dominant regimes do not really have a large impact (as the authors explicitly agree in line 133) on the main results of the study of determining transitions to more water-limited evaporation. Bear in mind that Figure 2 can therefore become a bit of a distraction when trying to state where regions are spending more time water-limited, especially when the main punch line of figure 4 is not clearly stated in these final paragraphs of the results.

Figure 4: This binary metric of migration is helpful. However, it would be more helpful to also know how much more time is being spent in the transitional regime as well, which is (I think) largely dictated by trends in soil moisture in the models. A figure that shows a percentage of time change would add more meaningful information on top of that of the more binary metric in figure 4.

L315: As a confidence test, the authors could repeat the analysis with AIC. BIC here may tend to select less parameterized models (those with less soil moisture breakpoints) while AIC may find more parameterized models. It is difficult to assess if this would change the main results in figure 4. This is optional given that there is no basis for arguing that AIC is better/worse than BIC. It just provides another sensitivity test for Figure 4 and how much regime classifications matter for the final results.

Reviewer #2 (Remarks to the Author):

Hsu et al., utilize the latest model simulations to identify possible changes in soil moisture control on latent heat flux globally under increasing CO₂ concentrations. Overall, this paper is well-written and methodologically sound. The authors classify 5 main regimes, but I think adding C100 is important, which has different underpinning drivers compared to C001. In addition, current analyses are only on "how soil moisture regimes change?". I think it is very critical to do more analyses to show "why soil moisture regimes change?" and "Why models differ a lot?" Only discussions are not sufficient.

Other Comments:

Abstract: Current version is too general. It is good to complement some region hotspots and adequate numbers.

Figure 2 and 4: It is really hard to identify squares or stars. Please correct this.

Lines 286-323: The Method is critical. Only citing references is not easy for readers. Please explicitly explain it.

Reviewer #1 (Remarks to the Author):

Hsu and Dirmeyer classify evaporative regimes using several climate models and determine that land surfaces are on average projected to be in more water-limited evaporative states. I find this to be an important and timely study. It is methodologically sound in using previously developed methods to identify evaporative regimes and having proper confidence tests – I would have done all the same steps myself and the authors answered all of my methodology questions in their text. I support the study, but have some doubts about how well ESMs at the CMIP level, which tend to have very simplistic representations of the land surface, can really capture EF-soil moisture relationships and soil moisture trends that drive the main results. See my major comment for more details. I acknowledge it is difficult for the authors to address this concern, but I recommend some ways they may be able to provide some confidence in the model results.

I do not wish to remain anonymous – Andrew Feldman.

We thank Dr. Feldman for the comments. Mainly, we have strengthened our motivation and more clearly set the frame of this study. As most of the additional analysis requested has already been provided either in our previous study or in the supplementary information, most of these comments are addressed by modification of the manuscript to make those connections clearly. We have added more description to the text discussing SM regime bias in our previous work, in which a comparison between models and observationally-based data had been made, and we provide a clearer linkage to the supplementary information, in which the spatial pattern of the time soil moisture spends in different regimes has been provided. We also checked results by using AIC and confirm that using AIC versus BIC for model selection only negligibly changes our results.

Major Comments

1) A major unanswered question and lynchpin of this study is how well CMIP climate models can really capture evaporation-soil moisture relationships as well as trends in soil moisture – both of which combine to give the results in Figure 4. Koster et al. 2006 in the GLACE experiments show a wide range of variability between models in this coupling that drives differences in the specific rain-soil moisture interactions they are looking at (which partly is a function of EF-soil moisture relationships). Short Gianotti et al. 2019 in their Figure 1 show a concerning wide range of variability of EF-soil moisture relationships in CMIP5 models that can often greatly differ from observations in the United States. Also see several studies by Jianzhi Dong and Wade Crow that show biases in land surface temperature – soil moisture coupling in models. Even though we can use an ensemble model average to communicate results here and find agreement (as in Figure 3), I am concerned with the high variance between models' EF-soil moisture relationships and L-A coupling (as shown in Koster et al. 2006 and Short Gianotti et al. 2019). Based on those studies, one could argue that we are still unsure how well

these climate models are really capturing land-atmosphere interactions given their large variance in the EF-soil moisture relationships. EF-soil moisture relationships are emergent behavior from many parameterizations (for example, plant-soil-climate interactions defining WP and CSM which are still highly uncertain in these global models), each having their own errors and differences between models. Additionally, precipitation projections are yet uncertain with heterogeneous changes that translate to uncertain changes in soil moisture trends globally as well (see studies by Alexis Berg). Ultimately, Figure 4 aggregates all of these uncertainties into how much more time is being spent in a water-limited evaporative stage in a changing climate.

Unless the authors argue differently that there have been many previous studies supporting how well climate models capture land surface evaporative regimes and soil moisture trends, I highly recommend that the authors devote more text to addressing these issues as a main shortcoming of the study that it is still not well known how well CMIP/climate models capture land-atmosphere coupling and regimes. Personally, I think we need more explicit observational studies that can be used to assess how well climate models are capturing water-limited evaporative regimes (Both the shapes of EF-soil moisture relationships and time spent in the water-limited regime as two example metrics). There are unfortunately not many observational, model-free options that can be used to test how well the ensemble model classifications in Figure 2 and current climate mean time spent in the transitional regime are capturing reality.

Beyond explicitly commenting on these limitations, I recommend that the authors also find a way to build confidence in the ensemble model results for the present climate runs. A few recommendations: (1) FLUXNET EF-soil moisture relationships can offer some sparse insights. (2) LST-soil moisture relationships from remote sensing (SMAP and MODIS) can give some larger spatial scale observational insights. (3) More careful selection of CMIP6 models by removing models that qualitatively show large deviations from expected nonlinear EF-soil moisture relationships in select test land domains (Figure 1 in Short Gianotti et al. 2019 can be a guide for this).

Short Gianotti, D.J., Rigden, A.J., Salvucci, G.D., Entekhabi, D., 2019. Satellite and Station Observations Demonstrate Water Availability's Effect on Continental-Scale Evaporative and Photosynthetic Land Surface Dynamics. *Water Resour. Res.* <https://doi.org/10.1029/2018WR023726>

Thank you for the detailed comment. We agree that it is needed to valid models' SM regimes with observations. As the reviewer has mentioned, pre-industrial simulations instead of historical simulations are used here, and thus comparing to the current climate is impractical – the models' historical simulations are appropriate for such a comparison. Nevertheless, we did compare the SM regime in historical simulations to reanalysis and SMAP L4 products in our previous

study (Hsu and Dirmeyer 2022). The difference was discussed in our original main text (Line 277-291). At this stage, reanalyses and SMAP L4 are the only products that meet global validation requirements and analyses have already been provided in the published literature. As a purely *in situ* set of observations with a long enough time series of daily SM and LE data (covering a few decades) is not available, we are not able to provide additional analysis here (as the reviewer has also mentioned). We admit that the description in the discussion section of original manuscript was not sufficient. To emphasize this issue more, we have added more description as in Line 291-297:

“Regarding these issues, a stricter validation of models’ SM regimes may become possible in the future, based on promising developments in global SM observations (e.g., Rodriguez-Fernandez et al. 2021; O and Orth 2021; Beck et al. 2021; Chaney et al. 2019; Seo and Dirmeyer 2022). A more comprehensive analysis of each component of SM-induced feedback in climate models will further help to evaluate the credibility of this study and to understand the causes of bias and change of models’ SM regimes.”

2) I found the motivation to be lacking a bit here. The authors could do a lot more to explain why we should care that land surface processes are tending to become more water-limited.

Thank you for the comment. We admit that the motivation was not clearly made in our original main text. We have integrated the Line Specific Comments L71 to addressing this comment. In the revision, we have added more description and cite more papers to clarify our scientific question, and to narrow down to our motivation as the following:

“Accordingly, SM regime transitions are good indicators of the shifting gears between modes of local SM:LE coupling that are an essential component of SM-induced feedback. Though improved understanding of how SM-induced feedback can change under global warming has been a priority for some time (Seneviratne et al. 2010), studies have focused more on quantifying the change of coupling in a climatological sense, such as a projected strengthening in land-atmosphere (Dirmeyer et al 2012, 2013; Berg et al. 2019&2021; Denissen et al. 2022) or an expansion in areas experiencing strong coupling (Seneviratne et al. 2006; Soares et al. 2019). However, climate in many locations is composed of days with and without active SM:LE coupling and thus the existence of SM-induced feedbacks. This suggests that under global warming, in addition to a stronger control of SM on LE, the climatological strengthening in coupling can also be attributed to a more frequent control of SM on LE, i.e. more days when SM is between WP and CSM, or even the situation where a transitional SM regime emerges locally when it did not exist previously. Thus, investigation on whether a warming climate leads to the shift, emergence, or disappearance of SM regimes, along with corresponding changes in the frequency of SM in each SM regime, has been lacking but is needed.

In this study, we determine the global patterns of existing SM regimes and their projected changes from state-of-the-art climate models. This enables quantification of how SM values migrate among dry, transitional, and wet

regimes due to global warming. Such diagnostic analyses can indicate which gear of SM-induced feedback is dominant at any location, and what changes may occur.” (Line 75-99)

Line Specific Comments

L55-57: Being rigorous, L55 “will not cause greater evaporation” and L57 “LE is also insensitive to SM variations” (and in caption in L625) are too definite of statements and may not be entirely correct. Soil moisture increases can still cause small evaporation increases in non-water limited regimes, but EF variations depend mainly on other factors.

Not required to reference but: Feldman et al. 2022 Fig. 6B+6C show with observations that there is still some energy flux sensitivity to soil moisture in the “wet regime” but it is just less. Dirmeyer et al. 2000 also shows this with models.

Dirmeyer, P.A., Zeng, F.J., Ducharne, A., Morrill, J.C., Koster, R.D., 2000. The Sensitivity of Surface Fluxes to Soil Water Content in Three Land Surface Schemes. J. Hydrometeorol. 1, 121-134. [https://doi.org/10.1175/1525-7541\(2000\)001<0121:TSOSFT>2.0.CO;2](https://doi.org/10.1175/1525-7541(2000)001<0121:TSOSFT>2.0.CO;2)

Feldman, A.F., Gianotti, D.J.S., Trigo, I.F., Salvucci, G.D., Entekhabi, D., 2022. Observed Landscape Responsiveness to Climate Forcing. Water Resour. Res. 58, e2021WR030316.

Thank you for pointing this out. We agree that the statement is not completely accurate. This is modified in the revised manuscript as “When SM>CSM, SM does not regulate evaporation (Dirmeyer et al. 2020; Feldman et al. 2022). Evaporation may even decline, as very wet soils correspond to periods of rainfall, cloudiness and limited sunshine.” (Line 56-58).

L71: A stronger, more explicit research question or objective is needed here. Stronger motivation is needed on why we should know the answer to this question and how well others have answered this previously. For example, Dirmeyer et al. 2012 referenced here have previously shown a projected increase in land atmosphere coupling strength with climate change which may or may not relate to the investigation here.

Thank you for the comment. We should have pointed out that an increase in coupling strength does not ensure a more moisture-limited world. Thus, examining the SM regime helps to address this issue. We admit that this argument was not clear in our original main text. While integrating the major comment #2 on motivation, this has been modified as the following:

“Accordingly, SM regime transitions are good indicators of the shifting gears between modes of local SM:LE coupling that are an essential component of SM-induced feedback. Though improved understanding of how SM-induced feedback can change under global warming has been a priority for some time (Seneviratne et al. 2010), studies have focused more on quantifying the change of coupling in a climatological sense, such as a projected strengthening in land-atmosphere (Dirmeyer et al 2012, 2013; Berg et al. 2019&2021; Denissen et

al. 2022) or an expansion in areas experiencing strong coupling (Seneviratne et al. 2006; Soares et al. 2019). However, climate in many locations is composed of days with and without active SM:LE coupling and thus the existence of SM-induced feedbacks. This suggests that under global warming, in addition to a stronger control of SM on LE, the climatological strengthening in coupling can also be attributed to a more frequent control of SM on LE, i.e. more days when SM is between WP and CSM, or even the situation where a transitional SM regime emerges locally when it did not exist previously. Thus, investigation on whether a warming climate leads to the shift, emergence, or disappearance of SM regimes, along with corresponding changes in the frequency of SM in each SM regime, has been lacking but is needed.

In this study, we determine the global patterns of existing SM regimes and their projected changes from state-of-the-art climate models. This enables quantification of how SM values migrate among dry, transitional, and wet regimes due to global warming. Such diagnostic analyses can indicate which gear of SM-induced feedback is dominant at any location, and what changes may occur.” (Line 75-99)

L75: Given my major comment, I think more motivation should be given for why the CMIP6 models are chosen. I am not a modeler so I'll word as a question: could we trust the analysis more if done with more land surface-focused models (CESM, CATCHMENT-CN) rather than ESM's that are coupled to a highly parametrized atmosphere and may have more complex land surface schemes (i.e., plant hydraulics) turned off?

Thank you for the question. We agree that analysis done with more complex land models would provide more insight. However, since our goal here is to obtain a model consensus result, model outputs from available CMIP6 data is the best choice, since their simulations have a consistent configuration as well as good (but not universal) availability of SM and LE data at daily time scale covering several decades. About the credibility between using CMIP6 and land surface-focused models, we are not able to provide a certain answer. Although how detailed a model represents physical processes does contribute to the credibility, a multi-model result can also enhance credibility from a statistical perspective. We believe opting for data availability to obtain a consensus result is the best choice for our study. To point out the motivation of selecting multiple CMIP6 models, we briefly describe the reason in the introduction.

“To examine these responses under warming with a climate model consensus perspective, daily data from eight climate models...” (Line 97)

Figure 2: As part of my comment below, it seems worth mentioning that there are very few pixels changing regime dominance based on this plot. 2.6% is very small. Perhaps shifts in the soil moisture distribution and specifically its mean are doing more of the work in changing time in the water-limited evaporative stage than changes in the tails of the soil moisture distribution that Figure 2 is probably more sensitive to.

Thank you for the comment. First, we have to correct the mistake that it should be 3.6% instead of 2.6% (Line 133 in modified main text). Then, we would like to clarify that this 3.6% means the areas with an emerging transitional regime. For areas with changing regime dominance, it is 13.3% by summing the grid cell areas marked with a star (where at least one regime appears or disappears). We provide this number in Line 116 "...this accounts for 13.3% of the global analyzed area." in the modified manuscript.

Finally, we would like to argue that 3.6% is not a small number since only 23.3% of the global area has no transitional regime in the pre-industrial climate, and thus $3.6/23.3\%=15\%$ increase in area where SM:LE becomes coupled under global warming. This argument is added in the revised manuscript:

"Only 23.3% of global land areas are identified as uncoupled (C001) for pre-industrial climate; a 3.6% expansion in global area with active SM:LE amounts to a shrinkage of 15% in uncoupled area." (Line 135-137)

We agree that examining the changing time in the water-limited evaporative stage is informative. Actually, this has been provided in the supplementary information (Fig S3). However, as the average of these patterns among climate models has large spread, due to inconsistent detection of WP and/or CSM among models, we only provide individual model results and not a consensus. As we prefer to focus on composite results in the main text, we chose to put these results in the supplementary information. In the modified text, we enhance the linkage between the main text and this individual result:

"Individual model changes are displayed in Fig.S3. Spread among climate models at any location can be large. This is mainly due to differences among models in the detection of WP and/or CSM. Nevertheless, the trend of how SM migrates in a changing climate demonstrates strong agreement in many regions." (Line 1714-178)

L141: For clarification, am I supposed to interpret Figure 4 as showing more common migration from dry to transitional (green) than from transitional to dry (red)? This could be made clearer in the text here. The main message of the study can get a little lost when starting with Figure 2, where the changes in the dominant regimes do not really have a large impact (as the authors explicitly agree in line 133) on the main results of the study of determining transitions to more water-limited evaporation. Bear in mind that Figure 2 can therefore become a bit of a distraction when trying to state where regions are spending more time water-limited, especially when the main punch line of figure 4 is not clearly stated in these final paragraphs of the results.

Thank you for the comment. We have added more description for interpreting Figure 4 in line 187-189:

"...and migration from dry to transitional (cyan) is the major tendency across the globe (42.3%)".

We also want to point out that Figure 2 instead of Figure 4 contains our main message as Figure 2 shows an expansion of certain SM regimes under warming, which inspires the title “coupling shifts into new gears”. Moreover, as both “the existence of a SM regime” and “the time SM spends in each regime” under warming are both rather unexplored topics. The former topic seems to be more large-frame, we believe Figure 4 plays a complementary role to Figure 2. To make this clearer, we additionally hint at the hypernym hierarchy of the results in our objective in the introduction:

“...Thus, investigation on whether a warming climate leads to the shift, emergence, or disappearance of SM regimes, along with corresponding changes in the frequency of SM in each SM regime, has been lacking but is needed.

In this study, we determine the global patterns of existing SM regimes and their projected changes from state-of-the-art climate models. This enables quantification of how SM values migrate among dry, transitional, and wet regimes due to global warming.” (Line 88-97)

Figure 4: This binary metric of migration is helpful. However, it would be more helpful to also know how much more time is being spent in the transitional regime as well, which is (I think) largely dictated by trends in soil moisture in the models. A figure that shows a percentage of time change would add more meaningful information on top of that of the more binary metric in figure 4.

Thank you for the comment. We agree that the migrating of SM among regimes can be attributed to the trend (also the expansion or narrowing of the distribution). We do quantify how the percentage of time in each SM regime changes, as shown by FigS3. As mentioned above (reviewer’s comment on Figure 2), we now link more clearly to this supplementary information in our main text and provide more discussion.

L315: As a confidence test, the authors could repeat the analysis with AIC. BIC here may tend to select less parameterized models (those with less soil moisture breakpoints) while AIC may find more parameterized models. It is difficult to assess if this would change the main results in figure 4. This is optional given that there is no basis for arguing that AIC is better/worse than BIC. It just provides another sensitivity test for Figure 4 and how much regime classifications matter for the final results.

Thank you for the comment. We have repeated the analysis selecting the models using AIC. The results are shown below. Comparing to the original result (Fig S2), we see only sporadic differences at a few locations with the different classification technique. For example, a few more green grid cells are found over the South Great Plains by using AIC. Overall, this indicates that penalty terms accounting for the number of parameters either in AIC or BIC does not substantially affect the regression selection. The extra check suggested by the reviewer solidifies our confidence in the test. We now state at Line 373:

“Akaike information criterion was also tested and found to produce nearly identical results.”

Reviewer #2 (Remarks to the Author):

Hsu et al., utilize the latest model simulations to identify possible changes in soil moisture control on latent heat flux globally under increasing CO₂ concentrations. Overall, this paper is well-written and methodologically sound. The authors classify 5 main regimes, but I think adding C100 is important, which has different underpinning drivers compared to C001. In addition, current analyses are only on “how soil moisture regimes change?”. I think it is very critical to do more analyses to show “why soil moisture regimes change?” and “Why models differ a lot?” Only discussions are not sufficient.

Thank you for the comment. C100 could provide a different perspective for determining SM regimes, and we initially sought to include it in our analysis. However, we would like to argue that defining C100 for this study is not practical due to the following reasons:

(1) In the previous study using the same SM regime detection method, a location was defined as C100 where the standard deviation of SM remains very close to zero regardless which regression best fits the SM-LE data. This was meant to classify the persistently dry desert areas as C100. However, for this study where we seek to determine the fraction of SM states in each SM regime, the definition is at odds with the notion of variable soil moisture. In reality and models in arid regions, daily SM can still exceed the WP or even CSM since sporadic rainfall events happen within a 50 year period, which trigger LE very effectively. Such a SM-LE coupling has also been evidenced in observations (e.g., Agam (Ninari) et al., 2004). Thus, we think forcing desert grid cells to be C100 is not practical or realistic.

Agam (Ninari), N., P.R. Berliner, A. Zangvil, E. Ben-Dor (2004) Soil water evaporation during the dry season in an arid zone, *J. Geophys. Res.*, 109.D16103, doi:10.1029/2004JD004802.

(2) C100 and C001 both indicate that a flat line best fits the SM-LE data. This means it requires an additional step to separate them, e.g., the standard deviation criterion described above to identify desert points. However, in our previous results, local SM-LE data that is best fitted by a flat line only occurred in rainforest, high latitude or high altitude areas – all energy-limited regimes. Thus setting an arbitrary criterion for separating C100 and C001 is ultimately unnecessary as all flat-line grid cells appear to be C001. This may have to do with model resolution – known areas that might fit C100 (e.g., the Atacama Desert) are not resolved by these climate models.

Determining the cause of differences among models is always a very difficult task. Every model inter-comparison project states this aim, and few accomplish much. The first step is to quantify the differences, which we do for these models and metrics. As this study already aims to diagnose SM regimes and their changes, examining the possible causes of the changes and the cause of differences among models is the next step, and beyond the scope of this paper. We think the results here are already adequate since they cover a rarely-explored topic for climate

change; we hope this work can trigger discussion and further investigation. In the modified manuscript, we have added more description to set the scope of this study and to provide more discussion potential topics of the following work:

“...Thus, investigation on whether a warming climate leads to the shift, emergence, or disappearance of SM regimes, along with corresponding changes in the frequency of SM in each SM regime, has been lacking but is needed.

In this study, we determine the global patterns of existing SM regimes and their projected changes from state-of-the-art climate models. This enables quantification of how SM values migrate among dry, transitional, and wet regimes due to global warming. Such diagnostic analyses can indicate which gear of SM-induced feedback is dominant at any location, and what changes may occur....” (Line 88-97)

“Regarding these issues, a stricter validation of models’ SM regimes may become possible in the future, based on promising developments in global SM observations (e.g., Rodriguez-Fernandez et al. 2021; O and Orth 2021; Beck et al. 2021; Chaney et al. 2019; Seo and Dirmeyer 2022). A more comprehensive analysis of each component of SM-induced feedback in climate models will further help to evaluate the credibility of this study and to understand the causes of bias and change of models’ SM regimes.” (Line 291-297)

Other Comments:

Abstract: Current version is too general. It is good to complement some region hotspots and adequate numbers.

We have attempted to accommodate the reviewer’s suggestion, but the abstract length is very restrictive: 150 words. We are already at that limit. We now quote the 15% reduction in wet regime (C001) and highlight the expansion of the transitional regime more clearly, including potential consequences. There is not space to be explicit about specific regional hotspots. The possible consequences are now also stated in the discussion section (Line 208-212):

“Furthermore, the expanding transitional regime could amplify local climate sensitivity to land-atmosphere feedbacks. This could make some regions more susceptible to unintended consequences from land management practices that alter soil moisture, such as irrigation, cropping choices, urbanization and water resource management.”

Figure 2 and 4: It is really hard to identify squares or stars. Please correct this.

Thank you for the comment. To make the plots clearer, we have modified the plot with more visible symbols and added to the caption.

For Figure 2, we have modified the legend symbols representing SM regimes under warming climate. Now, it is clearer that the star is always over the square (this also indicates that colored squares are everywhere). We also reinforce this point in the caption in Line 753 : "star color (over the square color) indicates a new candidate emerging in a warming climate at that grid cell."

For Figure 4, to clearly separate the grid cells with agreement > 30% and >60%, the symbol for agreement > 60% has been changed from a star to a large multiplication sign.

These modified plots are shown below:

Figure 2

Figure 4

Lines 286-323: The Method is critical. Only citing references is not easy for readers. Please explicitly explain it.

Thank you for the suggestion. Actually, the description we have provided in our original Methods Summary is a step-to-step workflow for attaining our results. We have also explained the motivation behind each step. The schematic plots displaying all the SM regime candidates have been provided in Figure 2. Only some description details of the mathematics is excluded in this manuscript. Since we were not clear about the best approach, we asked the Associate Editor, Dr. Efi Rousi, and she said the original Methods Summary was adequate. Replicating all those descriptions would essentially duplicate the content of our previously published paper (Hsu and Dirmeyer 2022; <https://doi.org/10.1175/JHM-D-21-0224.1>), potentially raising issues of self-plagiarism. As we have clearly cited this paper, we choose to preserve the original description in the Methods Summary.

REVIEWER COMMENTS

Reviewer #1 (Remarks to the Author):

The authors have done a great job answering comments. There is a much clearer motivation – nice job with lines 73-96 and lines 206-210. This is a great paper. I have a few minor comments remaining below.

Main Comment:

I may not have been clear in my major comment 1, but I think it would be helpful to lay out what drives the uncertainty in the computation of time spent in the transitional regime and its changes: the uncertainties lie in how well the CMIP models capture the regimes and their changes in shape/functional form as well as, importantly, the trends in the soil moisture distribution. I think this study should mention somewhere the uncertainty source related to how well these models capture the soil moisture distribution, in addition to the shape of the LE-soil moisture relationship shown in Fig. 1. The Hsu 2022 study is also helpful for providing confidence in the results. I also think it should be explicitly stated somewhere that these same CMIP models were compared against reanalysis and observation-based sources in this Hsu 2022 study to strengthen the results.

Minor Comment:

L20: "...growing the most." "Growing" is not clear here. I suggest making this more specific to its area such as "...growing the most in areal dominance."

L114: in the parenthesis "this" is ambiguous. Is the parentheses statement saying that 13.3% of pixels show a different regime detection in the future climate?

L286: spelling error. Should be "wetness"

Reviewer #2 (Remarks to the Author):

I appreciate the authors for their efforts in addressing my previous concerns. I agree with Reviewer1's point about the model uncertainties. I get a stronger feeling about this point when squinting at the discrepancy among models. For instance, in Figure 4, areas showing agreements of 5 in 8 models are very small, which really doubts the robustness of model projections in the aspect. I checked available experiments of 'picontrol' and '1pctCO2' from the current CMIP6 data pool, you have much more models to use now. I strongly suggest the authors to include them to build a larger ensemble. Ultimately, both the consensus and discrepancy parts should be highlighted. Then, readers can know which part we can trust more or be more careful about.

Last time, I suggest authors do more analyses to show "why soil moisture regime changes?" The authors do not want to investigate more on this. Whatever I would suggest authors at least provide an illustrative example and some discussions.

I suggest authors do a careful revision.

Other Comments:

Do you use total soil moisture (mrso) or only surface soil moisture in your analyses (mrsos)?

Lines 156-157: Depends on where you look at. Please provide a quantitate number on

this, maybe globally averaged.

Line 770: a small typo. Alphabet - O not number - 0.

Line 230-231: Reference?

Figure2: I have to say it is still unclear to me. Is the figure's resolution too low? Or Change the star color?

REVIEWER COMMENTS

Reviewer #1 (Remarks to the Author):

The authors have done a great job answering comments. There is a much clearer motivation – nice job with lines 73-96 and lines 206-210. This is a great paper. I have a few minor comments remaining below.

We thank Dr. Feldman for the comments. In the modified manuscript, we have included two additional models, GFDL-CM4 and INM-CM4-8, in the analysis. Please see the “Model Selection” section at the end for the process we used to define “available models”. We have updated the results throughout this paper; this includes all figures, tables, and the quantities we provide in the main text. The new results do not affect any of our interpretations but increase the robustness of the migration results (Fig 4).

Main Comment:

I may not have been clear in my major comment 1, but I think it would be helpful to lay out what drives the uncertainty in the computation of time spent in the transitional regime and its changes: the uncertainties lie in how well the CMIP models capture the regimes and their changes in shape/functional form as well as, importantly, the trends in the soil moisture distribution. I think this study should mention somewhere the uncertainty source related to how well these models capture the soil moisture distribution, in addition to the shape of the LE-soil moisture relationship shown in Fig. 1. The Hsu 2022 study is also helpful for providing confidence in the results. I also think it should be explicitly stated somewhere that these same CMIP models were compared against reanalysis and observation-based sources in this Hsu 2022 study to strengthen the results.

Thank you for the suggestion. In the modified manuscript, we have added this aspect when discussing the uncertainty of the results. This is as the following: “Moreover, different land models that inherently simulate different soil moisture distributions (Koster et al. 2009), which also often differ from observations, remains an issue (Hsu and Dirmeyer 2022). This leads to an essential discrepancy of preferred SM regime, in which a SM distribution states, among climate models and thus induces the uncertainty of the multi-model diagnosis of SM regimes and migrations in this study.” Lines 295-300

Minor Comment:

L20: “...growing the most.” “Growing” is not clear here. I suggest making this more specific to its area such as “...growing the most in areal dominance.”

Thank you for the comment. We have changed “growing” to “expanding” to make the statement clearer.

L114: in the parenthesis “this” is ambiguous. Is the parentheses statement saying that 13.3% of pixels show a different regime detection in the future climate?

Thank you for the comment. This means 13.3% of area (instead of pixels) show a different regime detected in the future climate. To clarify, this has been modified as:

“(crossed grid cells account for 13.9% of the global analyzed area)” – the percentage changed slightly with the inclusion of the extra models. Line 117

L286: spelling error. Should be “wetness”

Thank you for pointing this out. This is revised.

Reviewer #2 (Remarks to the Author):

I appreciate the authors for their efforts in addressing my previous concerns. I agree with Reviewer1's point about the model uncertainties. I get a stronger feeling about this point when squinting at the discrepancy among models. For instance, in Figure 4, areas showing agreements of 5 in 8 models are very small, which really doubts the robustness of model projections in the aspect. I checked available experiments of 'picontrol' and '1pctCO2' from the current CMIP6 data pool, you have much more models to use now.

I strongly suggest the authors to include them to build a larger ensemble. Ultimately, both the consensus and discrepancy parts should be highlighted. Then, readers can know which part we can trust more or be more careful about.

Thank you for the suggestion. In the modified manuscript, we have included two additional models, GFDL-CM4 and INM-CM4-8 in the analysis. Please see the "Model Selection" section at the end for the process we used to define "available models". We have updated the results throughout this paper; this includes all figures, tables, and the quantities we provide in the main text. The new results do not affect any of our interpretations but increase the robustness of the migration result (Fig 4). We also argue that agreements of 5 out of 8 models could be strong as this means that locally there are a maximum of seven categories, 5 models agreeing with the same category with so many choices is much less likely than, say, 5 of 8 coin flips agreeing. Now the comparison threshold is 7 out of 10 models, after the addition of two models. We also reinforce the discussion on discrepancy and highlight the regions we can trust more:

"Moreover, different land models that inherently simulate different soil moisture distributions (Koster et al. 2009), also often differ from observations, remains an issue (Hsu and Dirmeyer 2022). This leads to an essential discrepancy among preferred SM regimes in the climate models inducing uncertainty in the multi-model diagnoses and estimated migrations in this study" Lines 295-300

"Specifically, better consensus is reached over semi-arid regions (e.g. the Sahel, India, and Southern Great Plain) while discrepancies are relatively strong over rest of the globe (Fig.3a)." Lines 278-280

We already have several paragraphs to point out the areas that we should be concerned about (e.g. Line 281-290: about heatwave-active regions such as western North America; Line 255-269: about higher latitudes or higher altitudes). We believe such discussion, combined with the additions described above, is sufficient.

Last time, I suggest authors do more analyses to show "why soil moisture regime changes?" The authors do not want to investigate more on this. Whatever I would suggest authors at least provide an illustrative example and some discussions.

Thank you for the comment. We have looked back to our response, and we admit that the comment 'why soil moisture regime changes' was not properly addressed. There are two main sources contributing to soil moisture regime changes: (1) shifts in the soil moisture distribution and (2) shifts in critical soil moisture values (WP and CSM). Actually, a discussion of both factors has been provided in many places, but perhaps not coherently enough.

In Lines 232-253, we have demonstrated that the change in SM distribution is a contributor, and these changes could be ultimately attributed to changes in large scale circulation. This then is further connected to the "wet gets wetter, dry gets dryer" paradigm, which covers the integrated results that examine the causes of shifts in SM under climate warming. We have also pointed out that our results indicate that changes in CSM and WP are another contributor to soil moisture regime changes (recalling the relevant discussion in Lines 215-230). We also have summarized the causes in Lines 250-253.

In Lines 218-225, we have demonstrated that WP and CSM can be affected by environmental factors, respectively. Thus, we have presented the expected changes in WP and CSM under global warming and provided these results in Supplementary Fig 1. We have also discussed the properties of WP and CSM and provided speculation on causes of their changes in Line 218-225.

Additionally, we have tried to improve the descriptions by citing more relevant papers and mention more factors leading to change in SM distribution. The modifications are as the following: (This also addresses the comment on Lines 230-231 about citations).

“SM migration between regimes under global warming (Fig.4) shows a zonally consistent pattern. Such a response of SM distribution under warming is presumably due to changes in the large scale atmospheric circulation and/or regulated hydrological cycle (Lorenz and DeWeaver 2007; Byrne and O’Gorman 2015; Zhou et al. 2021).” Lines 232-234

“... critical SM values can shift under a warmer climate and thus parts of the SM spectrum can slide into different regimes without a significant change in the SM distribution. How precipitation, LE, and temperature individually affect the shifting gears of SM:LE coupling remains to be disentangled.” Lines 253-255

I suggest authors do a careful revision.

Other Comments:

Do you use total soil moisture (mrso) or only surface soil moisture in your analyses (mrsos)?

Thank you for the question. This was not clear. In the modified manuscript, we have put the details of the variables we use:
Daily SM (CMIP variable mrsos; soil water content in the top 10 cm) and LE (CMIP variable hfls) fields are taken from these simulations.” Line 327-328

Lines 156-157: Depends on where you look at. Please provide a quantitate number on this, maybe globally averaged.

Thank you for the comment. Lines 156-157 continues the discussion on the Sahara. As this was not clear, we modified this as “Over this area, detection of the wet regime is also inconsistent across models (Fig.3g) and the consensus is even less under warming (Fig.3h).” Line 154

Line 770: a small typo. Alphabet - O not number - 0.

Thank you for pointing this out. This is revised.

Line 230-231: Reference?

Thank you for the comment. This comment is integrated to the major comments (above) and addressed together.

Figure2: I have to say it is still unclear to me. Is the figure’s resolution too low? Or Change the star color?

Thank you for the comment. To make the plots clearer, we have improved the resolution of the figure. The location with new determined regimes is changed from a star to a large multiplication sign. We retain the 5-color scheme for the following reasons: (1) changing the star (now cross) colors will lead to 10 colors in the plot, losing the consistency established by the 5-color scheme (2) Using the same color scheme for pre-industrial and warming climate can clearly indicate an expansion or contraction of a regime. For example, yellow crosses are shown over the Amazon suggesting the transitional regime is expanding into the currently wet-only regime (green). The revamped figure is attached below.

Model selection

The list below is obtained from <https://esgf-node.llnl.gov/search/cmip6/> by selecting the following search parameters (website visited on 14th Oct 2022):

- (1) Experiment ID: piControl
- (2) Experiment ID: 1pctCO2
- (3) Table ID: day
- (4) Table ID: Eday
- (5) Variable: mrsos
- (6) Variable: hfls

Symbols in the table below:

v: new climate model data for consideration
 c: close relative to a model already used - not including
 d: inconsistent available ensemble member across variables
 m: data does not cover the required period (e.g. 1pctCO2 outputs covers only from 1st to 100th yr) or does not have the full set of required variables
 p: problems occurred when downloading (data server is unresponsive or the download request generates an empty file download shell)

Daily variables are stored in separate files. Thus, a source of model should have at least 4 files (the number in the bracket) as a necessary (but not necessarily sufficient) condition for availability of a complete set of data for our analysis. This filters out many climate models for consideration. Furthermore, the 1pctCO2 experiment must have years 100-150 of the simulation available. Variables from each experiment should be from the same ensemble member. Finally, we are not including climate models with close relatives (using only one, to best represent multi-model mode and spread). A climate model that passes all these requirements is included. Then, the downloading process can fail due to an unresponsive data server or generation of an empty file download shell. Many of these issues have persisted since our first attempt in early 2022. Eventually, GFDL-CM4 and INM-CM4-8 are the only two new models we have been able to add to the analysis. Green shading indicates the eight models that have already been used.

	piControl		1pctCO2		Final decision
	mrsos	hfls	mrsos	hfls	
ACCESS-CM2 (3)					
ACCESS-ESM1-5 (3)					
AWI-ESM-1-1-LR (4)					
CESM2 (2)					
CESM2-FV2 (1)					
CESM2-WACCM (3)					
CESM2-WACCM-FV2 (2)					
CMCC-CM2-HR4 (2)					
CMCC-CM2-SR5 (4)	v	v	m	m	m
CMCC-ESM2 (4)					
CNRM-CM6-1 (4)	v	v	v	v	c
CNRM-CM6-1-HR (4)					
CNRM-ESM2-1 (13)					
CanESM5 (16)					

EC-Earth3 (5)	v	v	x	x	m
EC-Earth3-AerChem (1)					
EC-Earth3-CC (3)					
EC-Earth3-LR (1)					
EC-Earth3-Veg (3)					
EC-Earth3-Veg-LR (3)					
FGOALS-f3-L (3)					
FGOALS-g3 (5)			v	v	m
GFDL-CM4 (8)	v	v	v	v	v
GFDL-ESM4 (2)					
HadGEM3-GC31-LL (10)	v	v	v	v	d
HadGEM3-GC31-MM (4)	v	v			m
ICON-ESM-LR (4)	v	v	v	v	p
IITM-ESM (4)					m
INM-CM4-8 (4)	v	v	v	v	v
INM-CM5-0 (4)					c
IPSL-CM5A2-INCA (2)					
IPSL-CM6A-LR (6)					
KACE-1-0-G (4)	v	v	v	v	p
MIROC-ES2L (4)	v	v	v	v	c
MIROC6 (4)					
MPI-ESM-1-2-HAM (6)	v	v	v	v	p
MPI-ESM1-2-HR (4)	v	v			m
MPI-ESM1-2-LR (6)			v	v	m
MRI-ESM2-0 (8)					
NESM3 (1)					
NorESM2-LM (6)	v	v	v	v	c
NorESM2-MM (6)					
TaiESM1 (2)					
UKESM1-0-LL (10)	v	v	v	v	p

REVIEWER COMMENTS

Reviewer #1 (Remarks to the Author):

The authors have adequately addressed my comments. I endorse the article for publication.

Reviewer #2 (Remarks to the Author):

I appreciate the authors for their efforts in addressing my previous concerns. I only have one major comment left. The authors now clarify that they utilize surface soil moisture (10cm) simulated from models to investigate soil moisture-evaporative coupling. This is a weakness of this study. Total soil moisture (mrso) is vital for many deep-rooted plants, such as forests, in controlling evaporation. Since this study is a model-based study, you can derive total soil moisture (mrso) easily. I suggest authors complement this analysis.

REVIEWER COMMENTS

Reviewer #1 (Remarks to the Author):

The authors have adequately addressed my comments. I endorse the article for publication.

We thank Reviewer for the nice discussion during the review process.

Reviewer #2 (Remarks to the Author):

I appreciate the authors for their efforts in addressing my previous concerns. I only have one major comment left. The authors now clarify that they utilize surface soil moisture (10cm) simulated from models to investigate soil moisture-evaporative coupling. This is a weakness of this study. Total soil moisture (mrso) is vital for many deep-rooted plants, such as forests, in controlling evaporation. Since this study is a model-based study, you can derive total soil moisture (mrso) easily. I suggest authors complement this analysis.

We thank reviewer for the suggestion. We have included analysis for total soil moisture by targeting the same set of climate models with identical experiments and the same period we analyzed for surface soil moisture, as has been indicated in Table S2. By the same filters for climate model selecting, only 4 climate models have the necessary data (MRI-ESM-0, CNRM-CM6-1-HR, INM-CM4-8, and MIROC6). Figure S4 shows that the solely wet regime area shrinks over tropics (001 -> 011). More areas are identified as solely transitional regimes when using total soil moisture and those areas are shrinking in many places except for INM-CM4-8. Furthermore, daily SM is mainly migrating into the transitional regime over most of the world, suggested by all 4 models (Figure S5). These additional results again suggest SM is shifting under warming climate. However, we cannot provide a credible conclusion as data from only 4 models are analyzed, and the results are not as clear as when surface soil moisture is used. Besides, the longer memory of sub-surface soil moisture content leads to a more weakly coupled daily SM:LE relationship, making the analysis here less physically reasonable. For these reasons, we provide these additional results as supplementary information. The figures are provided in the following page. The description of these results have been added to the method summary:

“We also provide the same analysis using total soil moisture content (CMIP variable mrso). Only 4 out of the 10 climate models (MRI-ESM-0, CNRM-CM6-1-HR, INM-CM4-8, and MIROC6) provide this variable for the same experiments and periods as for surface soil moisture data. These analyses suggest shrinkage in the solely wet regime area over the tropics (Figure S4) and an overall growth of the transitional regime (Figure S5). Despite lower credibility due to fewer analyzed climate models and less clarity in the patterns of change, these results also indicate shifts in the dominant SM regime under global warming.”
(Line 419-426)

Figure S4. As in Figure S2 but analysis is applied by using total soil moisture (mrsos) to replace surface soil moisture (mrsos) for the same set of experiments (Only data from MRI-ESM-0, CNRM-CM6-1-HR, INM-CM4-8, and MIROC6 were available at the time analysis was conducted).

Figure S5. As in Figure S3 for total soil moisture (mrso) instead of surface soil moisture (mrsos).